# Evaluation of Screening Program and Phylogenetic Analysis of SARS-CoV-2 Infections among Hospital Healthcare Workers in Liège, Belgium

**DOI:** 10.3390/v14061302

**Published:** 2022-06-14

**Authors:** Majdouline El Moussaoui, Nathalie Maes, Samuel L. Hong, Nicolas Lambert, Stéphanie Gofflot, Patricia Dellot, Yasmine Belhadj, Pascale Huynen, Marie-Pierre Hayette, Cécile Meex, Sébastien Bontems, Justine Defêche, Lode Godderis, Geert Molenberghs, Christelle Meuris, Maria Artesi, Keith Durkin, Souad Rahmouni, Céline Grégoire, Yves Beguin, Michel Moutschen, Simon Dellicour, Gilles Darcis

**Affiliations:** 1Department of Infectious Diseases and General Internal Medicine, University Hospital of Liège, 4000 Liege, Belgium; patricia.dellot@chuliege.be (P.D.); yasminebelhadj01@gmail.com (Y.B.); cmeuris@chuliege.be (C.M.); mmoutschen@chuliege.be (M.M.); gdarcis@chuliege.be (G.D.); 2Department of Biostatistics and Medico-Economic Information, University Hospital of Liège, 4000 Liege, Belgium; nmaes@chuliege.be; 3Department of Microbiology, Immunology and Transplantation, Rega Institute, Katholieke Universiteit Leuven, 3000 Leuven, Belgium; samuel.hong@kuleuven.be (S.L.H.); simon.dellicour@ulb.be (S.D.); 4Department of Neurology, University Hospital of Liège, 4000 Liege, Belgium; nicolas.lambert@chuliege.be; 5Department of Biothèque Hospitalo-Universitaire de Liège (BHUL), University Hospital of Liège, 4000 Liege, Belgium; stephanie.gofflot@chuliege.be; 6Department of Clinical Microbiology, University Hospital of Liège, 4000 Liege, Belgium; p.huynen@chuliege.be (P.H.); mphayette@chuliege.be (M.-P.H.); c.meex@chuliege.be (C.M.); sbontems@chuliege.be (S.B.); j.defeche@alumni.uliege.be (J.D.); 7Centre for Environment and Health, Department of Public Health and Primary Care, Katholieke Universiteit Leuven, 3000 Leuven, Belgium; lode.godderis@kuleuven.be; 8Institute for Biostatistics and Statistical Bioinformatics, Katholieke Universiteit Leuven, 3000 Leuven, Belgium; geert.molenberghs@uhasselt.be; 9Laboratory of Human Genetics, GIGA-Institute, University of Liège, 4000 Liege, Belgium; maria.artesi@uliege.be (M.A.); kdurkin@uliege.be (K.D.); 10Laboratory of Animal Genomics, GIGA-Medical Genomics, GIGA-Institute, University of Liège, 4000 Liege, Belgium; srahmouni@uliege.be; 11Department of Haematology, University Hospital of Liège, 4000 Liege, Belgium; celine.gregoire@chuliege.be (C.G.); yves.beguin@chuliege.be (Y.B.); 12Spatial Epidemiology Lab, Université Libre de Bruxelles, 1000 Brussels, Belgium

**Keywords:** COVID-19, SARS-CoV-2, healthcare workers, occupational exposure, infection prevention and control, healthcare-associated transmission, phylogenetic analysis

## Abstract

Healthcare workers (HCWs) are known to be at higher risk of developing severe acute respiratory syndrome coronavirus 2 (SARS-CoV-2) infections although whether these risks are equal across all occupational roles is uncertain. Identifying these risk factors and understand SARS-CoV-2 transmission pathways in healthcare settings are of high importance to achieve optimal protection measures. We aimed to investigate the implementation of a voluntary screening program for SARS-CoV-2 infections among hospital HCWs and to elucidate potential transmission pathways though phylogenetic analysis before the vaccination era. HCWs of the University Hospital of Liège, Belgium, were invited to participate in voluntary reverse transcriptase-polymerase chain reaction (RT-PCR) assays performed every week from April to December 2020. Phylogenetic analysis of SARS-CoV-2 genomes were performed for a subgroup of 45 HCWs. 5095 samples were collected from 703 HCWs. 212 test results were positive, 15 were indeterminate, and 4868 returned negative. 156 HCWs (22.2%) tested positive at least once during the study period. All SARS-CoV-2 test results returned negative for 547 HCWs (77.8%). Nurses (*p* < 0.05), paramedics (*p* < 0.05), and laboratory staff handling respiratory samples (*p* < 0.01) were at higher risk for being infected compared to the control non-patient facing group. Our phylogenetic analysis revealed that most positive samples corresponded to independent introduction events into the hospital. Our findings add to the growing evidence of differential risks of being infected among HCWs and support the need to implement appropriate protection measures based on each individual’s risk profile to guarantee the protection of both HCWs and patients. Furthermore, our phylogenetic investigations highlight that most positive samples correspond to distinct introduction events into the hospital.

## 1. Introduction

Over two years after the onset of the ongoing coronavirus disease 2019 (COVID-19) pandemic, the epidemiological situation is still a major concern in many parts of the world. Originally arising in the province of Wuhan (China), the severe acute respiratory syndrome coronavirus 2 (SARS-CoV-2) spread to Belgium, among other countries, through travelers returning from Tuscany, Italy [1]. It gave rise to the first wave of the disease, extending from March to June 2020 in Belgium, quickly followed by a second wave of infections, from September 2020 to February 2021 [2]. In many European countries, including Belgium, healthcare systems have been challenged to deal with this global crisis. During the first and the second epidemic waves, millions of people were confined to their homes by government decisions in order to minimize transmission of SARS-CoV-2 [3]. In contrast, healthcare workers (HCWs) were working at the frontline of the outbreak, which both directly and indirectly exposed them to infected patients or contaminated materials, putting them in turn at high risk of becoming infected [4,5,6,7,8,9]. Therefore, HCWs may be responsible for nosocomial outbreaks and may transmit SARS-CoV-2 to vulnerable patients. During previous SARS-CoV-1 (2003) and Middle East respiratory syndrome coronavirus (MERS-CoV, 2012) epidemics, nosocomial outbreaks were considered to play a crucial role in the amplification and spread of these viruses [10]. Concerning SARS-CoV-2, both importance of healthcare-associated transmission as a pandemic driving force and risk factors for HCWs to become infected are currently unclear [11,12,13,14,15,16]. Several studies investigating healthcare associated clusters among HCWs, through genomic and phylogenetic analysis, revealed a majority of community-acquired infections [11,17,18,19], whereas others [12,19,20,21,22,23,24,25,26] highlighted occupational transmissions such as patient-to-HCW and HCW-to-HCW. The contribution of community and healthcare associated transmission leading to SARS-CoV-2 infections among HCWs is still debated. The reported proportion of SARS-CoV-2 infections among HCWs is highly variable. Although still debated [11,27,28], there is an increasing body of evidence that HCWs are at higher risk for SARS-CoV-2 infection than general population [4,19,20,21,22,23,24,25]. Patient-facing HCWs including notably nurses, and HCWs working in COVID-19 units, showed the highest infection rates [4,6,7,8,10,16,29,30,31,32,33,34,35,36]. Protection of patients and HCWs from nosocomial SARS-CoV-2 infection is crucial for the control of the pandemic and justifies the implementation of protection measures including the use of appropriate personal protective equipment (PPE), isolation, hygiene, and effective ventilation, as well as rapid identification and isolation of infected patients and HCWs [37,38]. Another strategy being considered is the regular screening for SARS-CoV-2 infection of all HCWs through reverse transcriptase-polymerase chain reaction (RT-PCR) assays performed on respiratory samples (throat and nasopharyngeal swabs) [7,9,14,36]. In addition to self-isolation based on symptom recognition, regular screening of HCWs could further reduce transmission by identifying individuals with asymptomatic or presymptomatic infection [39,40]. To achieve optimal efficiency, this protective intervention should be modulated depending on the individual risk profile of being infected. It is crucial to understand transmission dynamics of healthcare-associated outbreaks, including the complex interplay between and respective role of HCWs in transmission, in order to inform infection prevention guidelines. Here, we evaluated the implementation of a voluntary screening program for SARS-CoV-2 infection among HCWs in a tertiary center using weekly RT-PCR assays during both the first and second epidemic waves in Belgium. We assessed the frequency of positive test results among HCWs and we evaluated the risk factors for infection among different occupational role categories. To identify possible transmission clusters, genome sequencing and phylogenetic analysis were performed on nasopharyngeal swabs or throat washes from a subgroup of the cohort. 

## 2. Materials and Methods

From April to December 2020, the University Hospital of Liège (Belgium) offered the opportunity for its staff, symptomatic or not, to carry out a SARS-CoV-2 test by RT-PCR. HCWs were invited to participate in voluntary nasopharyngeal swabs (NP) or throat washes (TW) RT-PCR testing every week [41]. Data from each HCW were collected through a questionnaire completed at the time of the first RT-PCR test, except when the subjects had also participated in a prospective study on the seroprevalence of anti-SARS-CoV-2 IgG antibodies at our institute [42]. In this case, data were collected through the same questionnaire that was completed between April and May 2020 for this previous study. The questionnaires covered background data on staff role and working area, whether the HCW was transferred to a working COVID-19 unit during the period study, whether the staff member wore a mask or not, information related to potential contacts with SARS-CoV-2 infected patients as well as demographic information including age, gender, height, weight, smoking history, comorbidities, and ongoing medical treatment.

Written informed consent was obtained from each participant and the study was approved by the Research Ethic Committee of the University Hospital of Liège (approval reference number: 2020:155, 7 May 2020).

### 2.1. Laboratory Assays

RT-PCR assays were routinely performed using different systems to detect two different SARS-CoV-2 target genes in respiratory samples: Cobas 6800 (Cobas^®^ SARS-CoV-2; Roche, Basel, Switzerland), Abbott m2000 (RealTime SARS-CoV-2, Abbott, Chicago, IL, USA), GeneXpert (Xpress SARS-CoV-2, Cepheid, Sunnyvale, CA, USA) or the SARS-CoV-2 N1 + N2 Assay (Qiagen, Hilden, Germany). All methods were calibrated using a quantified positive control provided by the Belgian National Reference Laboratory for SARS-CoV-2 (KUL Leuven, Leuven, Belgium). Samples were considered positive when the viral load detected was higher than or equal to one copy per milliliter (mL). RT-PCR results were considered as indeterminate when the samples were positive for one target gene and negative for the other. When several tests were performed in the same week for the same HCW, only the first positive result was included in the study. If there were no positive result, the first indeterminate result was included; if there were none, only one negative result was included.

### 2.2. SARS-CoV-2 Sequencing and Phylogenetic Analysis

Sequencing of SARS-CoV-2 genomes was performed for 45 samples. RNA extraction from nasopharyngeal swabs or throat washes (300 µL) was performed using a Maxwell 48 device and the Maxwell RSC Viral TNA kit (Promega) with a viral inactivation step using Proteinase K, following the manufacturer’s instructions. RNA was eluted in 50 µL of RNAse free water. 1.2 µL of SuperScript IV VILO^TM^ Master Mix (ThermoFisher Scientific, Waltham, MA, USA, ID 11756500) and 1.5 µL of H_2_O were combined with 3.3 µL of the eluted RNA to carry out Reverse Transcription, followed by incubation at 25 °C for 10 min, 50 °C for 10 min, and 85 °C for 5 min. PCR was carried out using Q5^®^ High-Fidelity DNA Polymerase (NEB) and primers to obtain 1200 bp amplicons as described by Freed and colleagues [43]. PCR conditions were set up according to the recommendations of the ARTIC Network sequencing protocol (https://artic.network/ncov-2019 (accessed on 1 May 2020). Samples were multiplexed following the manufacturer’s recommendations using the Oxford Nanopore Native Barcoding Expansion kits 1–12, 13–24, and 96 in conjunction with Ligation Sequencing Kit 109 (Oxford Nanopore, Oxford, UK). Sequencing was carried out on a Minion using R9.4.1 flow cells.

To investigate the evolutionary relationships among HCW infectious cases, phylogenetic analysis were performed based on an alignment made of (i) the 45 viral genomes obtained from the infections of HCWs and sequenced in the context of the present study, (ii) all Belgian sequences available on GISAID (www.gisaid.org (accessed on 1 March 2022)) and collected until 1 December 2020 (*n* = 3163), as well as (iii) all genomic sequences that were used in the European Nextstrain [44] build dating of 1 December 2020 (*n* = 3721). We first ran a maximum-likelihood phylogenetic analysis with the program IQ-TREE 2.0.3 3 [45] using a general time-reversible (GTR) nucleotide substitution model [46] with empirical base frequencies and four free site rate categories [47], which was selected as the optimal model using IQ-TREE’s ModelFinder tool. The phylogeny was then time-calibrated using the program TreeTime 0.8.4 5 [48].

Following a previously described analytical workflow [49,50], a discrete phylogeographic analysis was performed using the discrete diffusion model [51] implemented in the software package BEAST 1.10 [52], with the objective to identify independent introduction events of SARS-CoV-2 lineages into the hospital. Specifically, the time-scaled phylogenetic tree was used as a fixed empirical tree and only considered two possible ancestral locations: “hospital” and “other location”. We conducted Bayesian inference through Markov chain Monte Carlo (MCMC) for 3 × 10^5^ iterations and sampled every 1000 iterations. We inspected MCMC convergence and mixing properties using the program Tracer 1.7 [53] to ensure that effective sample size (ESS) values associated with estimated parameters were all higher than 200. After having discarded 10% of sampled trees as burn-in, a maximum clade credibility (MCC) tree was generated using the program TreeAnnotator 1.10 [52], and then the resulting MCC tree were used to delineate phylogenetic clades corresponding to independent introduction events into the hospital.

### 2.3. Statistical Analysis

Quantitative variables were presented as means and standard deviations (SD) or median (Q1–Q3) while frequency tables (numbers and percentages) were used for qualitative variables. Univariate logistic regression analysis was performed to evaluate the impact of the demographic characteristics and HCWs staff role on the risk of presenting at least one RT-PCR positive result. The results were reported as odds ratios (OR), 95% confidence Interval (95% CI) and *p*-values. HCWs were grouped into seven categories according to their staff role (administrative staff, laboratory staff, physicians, paramedics, nurses, research scientists, and technicians). Administrative staff and research scientists were regrouped under the term “non-patient facing group”. Since their role does not require close contact with patients or the hospital environment and many of them worked in an off-site location separate from hospital sites, this non-patient facing group was used as a control for the present analysis. The longitudinal aspect was also studied using generalized estimating equations (GEE) modeling RT-PCR results according to demographic characteristics, HCWs staff role, and time since beginning of the study. Statistically significant variables in univariate models were included in a multivariate GEE model. Adherence to the study protocol was evaluated through the comparison of the number of weeks of participation using linear regression models. Results were reported using estimated coefficient ± standard error (SE) and *p*-values. Missing data were not replaced, and calculations were always performed on the maximum amount of available data. A *p*-value was considered statistically significant if less than 0.05. Data analysis was carried out using SAS software (version 9.4) for Windows. The R package ggplot2 (version 3.6.1) was used for the figures.

## 3. Results

### 3.1. Characteristics of HCW Groups

During the observation period (April to December 2020), 846 HCWs were tested weekly for SARS-CoV-2 infection using RT-PCR assays. Among them, 143 subjects were excluded from the study because they did not complete the requested questionnaire; thus, no information was available to conduct the study (Figure 1). In total, nearly one in ten employees (703/6263) of our institution who participated in the SARS-CoV-2 testing campaign during the study period completed a questionnaire and were therefore included in this study. Demographic characteristics of the HCWs cohort are presented in Table 1. The average age of individuals in the cohort was 41.4 years (SD ± 11.3 years) and the average body mass index was 24.3 kg/m^2^. The cohort was skewed towards females, with only 20% of the cohort comprising males. Comorbidity information was available for the 661 HCWs. Among them, 179 (27.1%) had at least one of the comorbidities currently considered as risk factors for severe COVID-19 (including diabetes mellitus, hypertension, cardiovascular disease, stroke, liver failure or cirrhosis, renal failure, chronic lung disease, asthma, immunodeficiency, and cancer), and 75 (11.3%) were smokers. Nurses represented 23.4% of the cohort, administrative staff 22.1%, laboratory staff 19.8%, paramedical 17.1%, physicians 10.8%, technicians 5.3%, and research scientists 1.4%. Nurses, physicians, and paramedics were the groups most exposed to patients, with 96%, 85%, and 66% of them being in contact with patients, respectively. Among the laboratory staff, 32.4% handled potentially contaminated respiratory samples.

### 3.2. SARS-CoV-2 PCR Testing

During the study period, our laboratory processed and provided SARS-CoV-2 RT-PCR results for 5411 samples collected from 703 included individuals. Among these, 316 samples were excluded from the analysis for one of the following reasons: no available results (37 samples), duplicate tests (223 samples), ineligible sampling method (14 saliva samples), or because the same participant was sampled several times during the same week (42 samples; Figure 1). 79 samples were obtained via NP swabs and 5016 via TW. The number of samples per individual ranged from 1 to 28, with an average of 7.2 (±5.4) tests per subject (Appendix A) and the average duration of participation was 14 (±11) weeks. Although HCWs were offered weekly RT-PCR testing, the workers attended less frequently than that with an average interval between RT-PCR tests of 2.2 (±2.0) weeks. Adherence was significantly lower in HCWs with a previously positive test result (*p* = 0.004). Overall, adherence to the study protocol was low and varied notably according to the occupational role of the staff. Females (2.1 ± 1.0, *p* = 0.033), nurses (7.1 ± 1.1, *p* < 0.001), and HCWs working in a direct contact with patients (3.6 ± 0.81, *p* < 0.001) were the most adherent subgroups (Table 2). In total, 212 SARS-CoV-2 RT-PCR assays were positive, 15 were indeterminate and 4868 returned negative (Figure 1). Of the 703 included subjects, 156 (22.2%) presented with at least one positive or indeterminate test result during the study period. For the remaining 547 participants (77.8%), all SARS-CoV-2 tests were negative (Figure 1). The respective peaks of RT-PCR assays performed, and the positive results occurred concomitantly between the 40th and 52nd weeks of the year 2020 (from September 28 to December 27, 2020), corresponding to the second wave of the COVID-19 epidemic in Belgium (Figure 2a,b).

### 3.3. Association of HCWs Role with SARS-CoV-2 Infection

Laboratory staff handling respiratory samples (OR (95% CI): 2.2 (1.1–4.8); *p* = 0.004), paramedics (OR (95% CI): 2.0 (1.1–3.5); *p* = 0.020) and nurses (OR (95% CI): 1.9 (1.1–3.3); *p* = 0.015) were at higher risk for SARS-CoV-2 infection (with at least one RT-PCR test positive) compared to the control non-patient facing group (Figure 3, Table 3). The test positivity rates increased over time as the study progressed (*p* < 0.001; Appendix A). We found no significant association between SARS-CoV-2 infection and demographic characteristics (Table 3). The longitudinal aspect studied using GEE led to the same conclusions (Appendix A).

### 3.4. Viral Sequencing and Phylogenetic Analyses

Our phylogenetic analysis revealed that most positive samples from HCWs that sequenced in the context of the present study corresponded to independent SARS-CoV-2 introduction events into the hospital. Specifically, we identified a minimum of 35 introduction events into the hospital (95% highest posterior density interval = [36–38]) for 45 sequences sampled among HCWs from the hospital (Figure 4 and Appendix A). We estimated that 30 (95% highest posterior density interval = [27–32]) out of 45 sequenced HCW positive cases were not directly related to any other HCW sample analysed in our study. We only identified five (95% highest posterior density interval = [4–6]) pairs of samples that seem to be directly related to each other, as well as two infectious clusters likely connecting four and eight samples, respectively.

## 4. Discussion

Through the rapid establishment of an expanded in-hospital HCWs SARS-CoV-2 screening program using voluntary RT-PCR assays, we observed that almost a quarter (22.2%) of the HCWs study cohort were tested positive for SARS-CoV-2 infection between March and December 2020. Almost all infections occurred between the end of September and December 2020, corresponding to the second epidemic wave in Belgium. Several factors could explain this observation. First, in contrast to other locations more affected by material shortages, all staff members working in the COVID-19 units of our hospital had access to PPE and other protective materials since the beginning of the pandemic (Figure 2). Second, most of the tests performed during our voluntary screening program were conducted during the second wave of the epidemic (Figure 2a). Then, staff exposure to SARS-CoV-2 outside the workplace may have varied between the different epidemic waves, which could for example be related to the Belgian government adopting less stringent social restrictive measures during the second epidemic wave. Moreover, to overcome the problem of healthcare personnel staffing shortages during the second epidemic wave, HCWs with COVID-19 could continue to work in healthcare facilities, including in non-COVID wards, which might have favored transmission to other HCWs. Finally, Belgium was particularly affected by this second epidemic wave, and Liège was even considered as the epicenter of the pandemic in Europe at that time.

RT-PCR assay positivity rates varied depending on the occupational role of the HCW included in the study and were significantly higher for nurses, paramedics and laboratory staff handling respiratory samples compared to the non-patient-facing control group, as already suggested in other studies [4,6,7,8,10,16,29,30,31,32,33,34,35,36]. Although we cannot formally exclude the contribution of infections transmitted outside the hospital, our observations suggest that direct contact with infected patients or contaminated materials is a risk factor for infection. In this context, our phylogenetic investigations highlight that most yet not all positive cases among HCWs corresponded to distinct introduction events into the hospital. Our results support the role of community acquired infections in HCWs, who may then introduce the virus into the facility, and are in line with most studies investigating dynamics SARS-CoV-2 transmission among hospital employees [19,22,28,55,56]. Interestingly, physicians were not at an increased risk of infection compared to the control group. One possible explanation for this observation is that their contacts with patients and coworkers were less close and briefer than those of nurses and paramedics. During the pandemic, physician-patient interactions were rethought and reorganized in order to limit close contact as much as possible [57]. 

Many studies support the widespread adoption of iterative screening strategies for all HCWs, assuming that presymptomatic or asymptomatic SARS-CoV-2 carriers might significantly contribute to COVID-19 outbreaks [39,40,58]. However, there is currently no recommendation because of the weak evidence of transmission dynamics, particularly using genomic sequencing [14,59], and the role of HCWs in initiating or amplifying nosocomial outbreaks remains unclear. One possibility in order to make screening strategies as efficient as possible would be to focus screening programs on HCWs at higher risk of being infected as nurses, paramedics, and laboratory staff handling respiratory samples. This approach will enable infected HCWs to self-isolate at the time of peak infectivity [60] and prevent uncontrolled staff-to-staff or staff-to-patient transmission, which could lead to substantial morbidity and mortality in a particularly vulnerable patient group [61]. Moreover, such strategies might have potentially positive effects on the mental health of HCWs. HCWs reported high levels of psychological distress, including fear of infecting themselves and their environment [62]. This fear is even more pertinent to HCWs in contact with infected patients or contaminated materials. In previous epidemics, HCWs screening programs have boosted morale, decreased absenteeism, and potentially reduced long-term psychological sequelae [63]. The screening protocol enables HCWs to return to work more rapidly and might have an additional positive effect on health behavior [9,13].

The number of tests performed varied according to the occupational role profile, which is in line with the results reported by Modenese and colleagues [57] and Jones and colleagues [64]. HCWs were enrolled in a voluntary testing program with a flexible follow-up schedule, which led to different attendance frequencies. Indeed, women, HCWs patient-facing groups, and most notably nurses, were subpopulations demonstrating higher attendance rates, further supporting that they may represent the most suitable population for iterative screening strategies. 

Several elements still need to be determined before implementing of such screening strategies in daily practice. A recent study suggested the need for weekly testing to prevent 16 to 33% of onward transmission from HCWs [15,65], while others proposed screening every 2 to 4 weeks [64]. Therefore, the optimal testing frequency should be further studied. Then, the sampling method should also be adjusted to increase compliance. In this respect, our team has recently reported that alternative specimen sampling techniques, such as throat wash, should be considered to improve SARS-CoV-2 testing strategies in HCWs [41]. Although NP swabs remain a more sensitive collection method than TW when performed early after the first symptom onset, compliance is better among HCWs with this alternative method because NP swabs are more invasive and require a second person for collection.

We must acknowledge several limitations to our study. Firstly, the conclusions drawn from our data should consider the overall low adherence of the participants to the screening protocol. Secondly, the collected data did not allow us to distinguish laboratory staff who handled respiratory samples in a laminar flow hood from those who did not. Therefore, the differential risk between these two populations could not be evaluated. Moreover, we were not able to sequence all genomes from all positive samples. Thus, we may have missed potential clusters. Finally, we focused on phylogenic analysis to detect clusters. However, epidemiological data including contact tracing investigations are also necessary to confirm the existence of clusters. 

## 5. Conclusions

In conclusion, we evaluated the establishment of a voluntary SARS-CoV-2 infection screening program for HCWs in a tertiary center during first and second COVID-19 epidemic waves. This approach identified differential risk of becoming infected depending on the occupational role, with nurses, paramedics, and laboratory staff handling respiratory samples found to be at higher risk when compared to the non-patient facing control group. Moreover, HCWs in contact with patients, most notably nurses, were more likely to adhere to screening protocol. Therefore, our data suggest that these HCWs may represent the most suitable population for iterative targeted screening strategies. However, several elements need to be determined to make this strategy as efficient as possible, such as the optimal testing frequency or optimal sampling method. In addition, our phylogenetic analysis indicates that most positive HCW samples correspond to the introduction of distinct transmission chains into the hospital. Finally, our results further support the need to implement appropriate protection measures based on each individual’s risk profile to guarantee the protection of both HCWs and patients. Even if the vaccination campaign has now greatly modified the scenario of the COVID-19 pandemic, including among HCWs, our study provides data that can be useful for further development of strategies to mitigate the occupational risk of becoming infected, notably by the new SARS-CoV-2 variants, and therefore, the evolution of the pandemic.

## Figures and Tables

**Figure 1 viruses-14-01302-f001:**
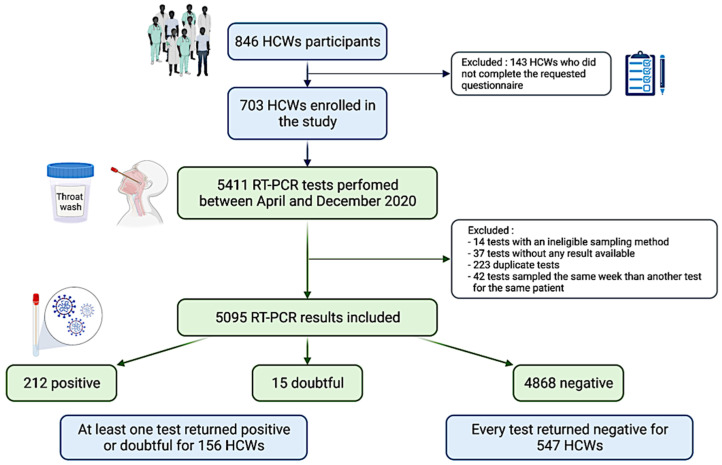
Study design. Among 846 consenting healthcare workers (HCWs) participants, 143 subjects were excluded from the study because they did not complete the requested questionnaire. Between April and December 2020, 5411 tests were performed in the 703 individuals included in the study. Among these tests, 316 were excluded for one of the following reasons: no results available (37 samples), duplicate tests (223 samples), ineligible sampling method (14 saliva samples), and two samples performed the same week for the same patient (42 samples). 212 SARS-CoV-2 reverse transcriptase-polymerase chain reaction (RT-PCR) assays were positive, 15 were indeterminate and 4868 returned negative. Of the 703 included subjects, 156 presented with at least one positive or indeterminate test result during the study period. For the remaining 547 participants, all SARS-CoV-2 tests were negative. Reprinted with permission from ref. [54].

**Figure 2 viruses-14-01302-f002:**
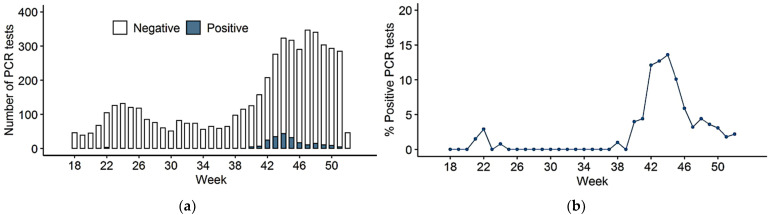
(**a**) Evolution of the number of reverse transcriptase-polymerase chain reaction (RT-PCR) assays performed and their results over time; (**b**) SARS-CoV-2 RT-PCR positive results rates (%) over time.

**Figure 3 viruses-14-01302-f003:**
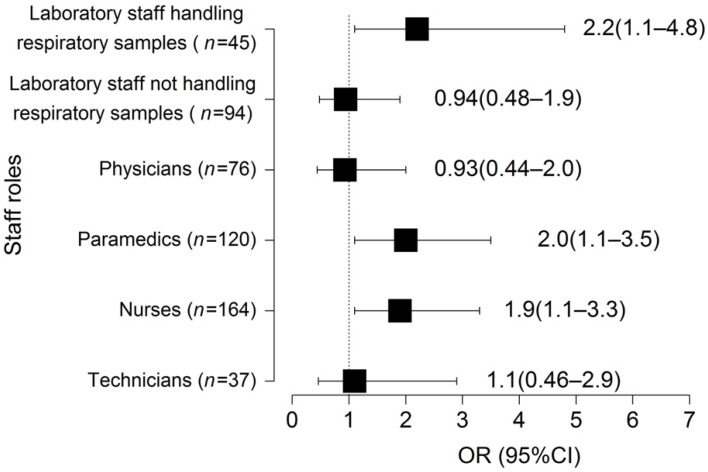
Impact of the healthcare workers (HCWs) staff role on the risk of developing SARS-CoV-2 infection (at least one RT-PCR positive result). Odds ratio and 95% confidence intervals calculated by logistic regression when compared to control non-patients facing group (administrative staff and research scientists).

**Figure 4 viruses-14-01302-f004:**
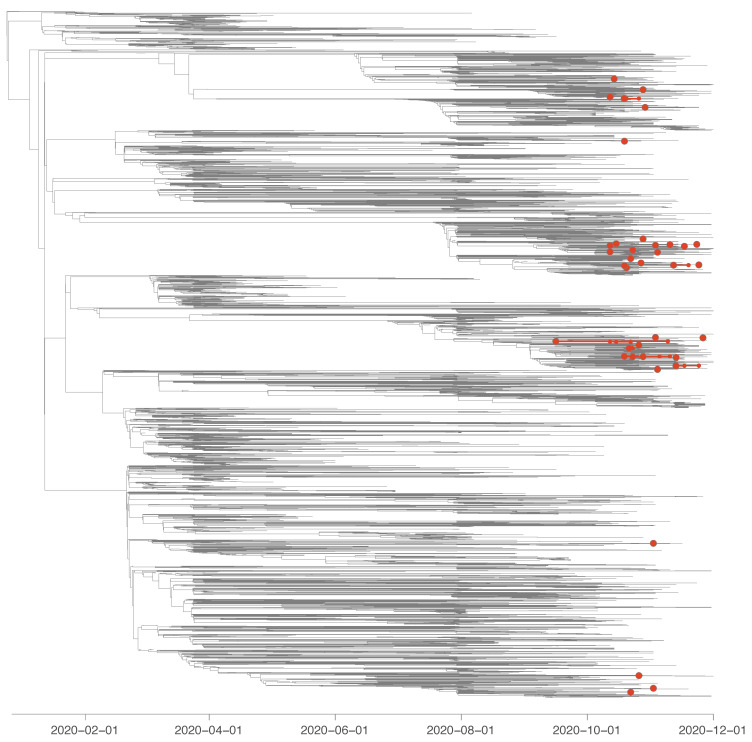
Time-scaled phylogeny in which we identified phylogenetic clades introduced in the University Hospital of Liège (Belgium) and delineated through a discrete phylogeographic reconstruction along the tree (while only considering two potential ancestral locations: “hospital” and “other location”). We identified a minimum of 35 introduction events into the hospital (95% highest posterior density interval = [36–38]) for 45 sequences sampled among healthcare workers (HCWs) from the hospital. On the phylogeny, large red nodes correspond to the most ancestral node of each clade resulting from an introduction event into the hospital. Most of these clades consist of only one sampled sequence: 30 (95% highest posterior density interval = [27–32]) out of 45 sequenced positive cases corresponded to independent introduction events into the hospital. In the figure, small red nodes correspond to sampled sequences that would not result from a distinct introduction event into the hospital. In other words, smaller red nodes are tip nodes belonging to clades gathering at least two sequences sampled among HCWs from the hospital. In the figure, smaller red nodes are tip nodes corresponding to sequences sampled among HCWs but that do not result from a distinct introduction event into the hospital. In other words, smaller red nodes correspond to clades gathering at least two sequences sampled among HCWs, and the phylogenetic branches of these clades are also highlighted in red. See Appendix A for an alternative circular visualization of this annotated phylogenetic tree.

**Table 1 viruses-14-01302-t001:** Characteristics of the healthcare workers (HCWs) study cohort.

Characteristics	Data Available (*n* = 703)	Results ^1^
Demographics		
Age (years)	703	41.4 ± 11.3
Female Gender	703	560 (79.7)
Height (cm)	661	168 ± 9
Weight (kg)	659	68.7 ± 13.5
BMI (kg/m^2^)	659	24.3 ± 4.2
Smokers	661	75 (11.3)
Comorbidities		
Diabetes mellitus	661	25 (3.8)
Hypertension	661	48 (7.3)
Heart failure/coronary artery disease	661	6 (0.9)
Stroke	661	1 (0.1)
Liver failure/cirrhosis	661	1 (0.1)
Renal Failure	661	1 (0.1)
Chronic lung disease	661	3 (0.4)
Asthma	661	70 (10.6)
Autoimmune disease	661	50 (7.6)
Immunodeficiency	661	6 (0.9)
Hematological cancer	661	3 (0.4)
Non hematological cancer	661	18 (2.7)
Organ or cell transplantation	661	0 (0.0)
Taking medication	661	438 (66.3)
Staff role		
Administrative staff	701	155 (22.1)
Laboratory staff	701	139 (19.8
Handling respiratory samples	701	45 (6.4)
Physicians	701	76 (10.8)
Paramedics	701	120 (17.1)
Nurses	701	164 (23.4)
Research scientists	701	10 (1.4)
Technicians	701	37 (5.3)
In contact with patients	703	395 (56.2)

^1^ Results are mean ± SD or *n* (%) as appropriate.

**Table 2 viruses-14-01302-t002:** Adherence to the study protocol.

Characteristics	*n*	Number of Weeks of Participation ^1^	Comparison (Coef. ± SE, *p*-Value)
Age (years)			
20–29	139	9 (3–21)	0.058 ± 0.036, *p* = 0.11
30–39	208	10 (5–26)	
40–49	177	11 (5–25)	
≥50	179	12 (6–26)	
Gender			
Female	560	11 (5–25)	2.1 ± 1.0, *p* = 0.033
Male (reference)	143	9 (4–21)	
Staff role			
Administrative staff (reference)	155	8 (4–24)	-
Laboratory staff	139	10 (5–22)	0.82 ± 1.2, *p* = 0.49
Physicians	76	12 (4–25)	1.3 ± 1.4, *p* = 0.35
Paramedics	120	8 (5–15)	−1.7 ± 1.2, 0.17
Nurses	164	24 (8–30)	7.1 ± 1.1, *p* < 0.0001
Research scientists	10	5 (1–16)	−5.3 ± 3.4, *p* = 0.12
Technicians	37	7 (2–21)	−1.1 ± 1.9, *p* = 0.55
In contact with patients			
Yes	395	13 (5–27)	3.6 ± 0.81, *p* < 0.0001
No (reference)	308	8 (5–21)	-

^1^ Results are Median (Q1–Q3) and estimated coefficient ± Standard Error (SE), *p*-value linear regression.

**Table 3 viruses-14-01302-t003:** Impact of the demographic characteristics and healthcare workers (HCWs) staff role on the risk of presenting at least one reverse transcriptase-polymerase chain reaction (RT-PCR) positive result. Adjusted odds ratio and 95% confidence intervals calculated by logistic regression when compared to control non-patients facing group (administrative staff and research scientists).

Characteristics	All Negative RT-PCR Results (*n*= 547)		At Least One Positive RT-PCR Result (*n* = 156)		Logistic Regression Models	
	*n* Non Missing	Result ^1^	*n* Non Missing	Result ^1^	OR (95% CI)	*p*-Value
Demographics						
Age (years)	547	41.4 ± 11.4	156	41.7 ± 11.2	1.0 (0.99–1.02)	0.79
Gender, women	547	432 (79.0)	156	128 (82.0)	0.82 (0.52–1.3)	0.40
Heigth (cm)	514	168 ± 9	147	168 ± 9	1.0 (0.98–1.02)	0.91
Weigth (kg)	512	68.6 ± 13.7	147	69.3 ± 13.0	1.0 (0.99–1.02)	0.59
BMI (kg/m^2^)	512	24.3 ± 4.2	147	24.6 ± 4.3	1.0 (0.97–1.1)	0.51
Smoking	514	64 (12.4)	147	11 (7.5)	0.57 (0.29–1.1)	0.098
Comorbidities						
Diabetes mellitus	514	21 (4.1)	147	4 (2.7)	0.66 (0.22–2.0)	0.45
Hypertension	514	37 (7.2)	147	11 (7.5)	1.0 (0.52–2.1)	0.91
Heart failure/coronary artery disease	514	5 (1.0)	147	1 (0.7)	0.70 (0.10–6.0)	0.74
Stroke	514	0 (0.0)	147	1 (0.7)	-	-
Liver failure/cirrhosis	514	1 (0.2)	147	0 (0.0)	-	-
Renal failure	514	1 (0.2)	147	0 (0.0)	-	-
Chronic lung disease	514	3 (0.6)	147	0 (0.0)	-	-
Asthma	514	52 (10.1)	147	18 (12.2)	1.2 (0.70–2.2)	0.46
Autoimmune disease	514	43 (8.4)	147	7 (4.8)	0.55 (0.24–1.2)	0.15
Immunodeficiency	514	5 (1.0)	147	1 (0.7)	0.70 (0.10–6.0)	0.74
Hematological cancer	514	3 (0.6)	147	0 (0.0)	-	-
Non hematological cancer	514	14 (2.7)	147	4 (2.7)	1.0 (0.32–3.1)	1.0
Staff role	545		156			
Control group: administrative staff and research scientists		137 (25.1)		28 (18.0)	-	-
Laboratory staff handling respiratory samples		31 (5.7)		14 (9.0)	2.2 (1.1–4.8)	0.0035
Laboratory staff not handling respiratory samples		79 (14.5)		15 (9.6)	0.94 (0.48–1.9)	0.87
Physicians		64 (11.7)		12 (7.7)	0.93 (0.44–2.0)	0.85
Paramedics		86 (15.8)		34 (21.8)	2.0 (1.1–3.5)	0.020
Nurses		118 (21.7)		46 (29.5)	1.9 (1.1–3.3)	0.015
Technicians		30 (5.5)		7 (4.5)	1.1 (0.46–2.9)	0.75
In contact with patients	547	296 (54.1)	156	99 (63.5)	1.5 (1.02–2.1)	0.039

^1^ Mean ± SD or *n*(%).

## Data Availability

The data presented in this study are available on request from the corresponding author. The data are not publicly available due to restrictions privacy.

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
