# Peer review of "Evaluation of Screening Program and Phylogenetic Analysis of SARS-CoV-2 Infections among Hospital Healthcare Workers in Liège, Belgium"

_viruses, 2022, doi:10.3390/v14061302_

Round 1
Reviewer 1 Report
This paper investigate the implementation of voluntary screening program for SARS-CoV-2 infections among 40 hospital HCWs and elucidate potential transmission pathways though phylogenetic analysis. This study is of interest for pertinent readers, researchers and practitioners. Therefore, I am of opinion that this study could be a good inclusion to the existing literature. However, manuscript requires some improvements. The authors are advised to consider following comments for the improvement.
- There's no need to provide headings i.e., background, methods results etc. Kindly write abstract without them.
- Introduction lacks detailed literature review. Please concisely discuss outcomes of similar studies.
- Please provide description of GEE for more clarity.
- Fig 2b indicates that real increase in positive cases occurred after 38th week. However, the authors didn't provide good discussion on relationship between time and surge in positive cases. Please discuss this aspect.
- Please provide results of statistical analysis/tests performed in this study. Currently it is difficult to assess the appropriateness of results of the current study. Provide a detailed Table and relevant discussion.
- The content of questionnaire is also unclear. Please provide detail of questionnaire used in this study.
- In conclusion, findings of this study are very important and interesting. However, comparison must be drawn between findings of the current study and other similar studies.
Reviewer 2 Report
* Comments and Suggestions for Authors
The manuscript “Evaluation of screening program and phylogenetic analysis of SARS-CoV-2 infections among hospital healthcare workers in Liège, Belgium” by Moussaoui et al. is a well written, and interesting study evaluating a screening program for SARS-CoV-2 in health care workers in a tertiary care center in Belgium. Both the study methodology and conclusions drawn from the outcomes are straightforward and concise.
There are only minor revisions required to be addressed:
Abstract: Please remove the numbers and sections from the abstract. (1) Background: (2) Methods: (3) Results: (4) Conclusion:
The abstract reads well on its own without the sections.
Line 38: grammar should be ‘identifying’ and ‘understanding’
Line 41: ‘..investigate the implementation of a voluntary screening..’
Line 46: Please use the more scientifically correct terms ‘indeterminate’ or ‘inconclusive’ instead of ‘doubtful’. There are a few more instances of the word ‘doubtful’ in the manuscript, which should be changed to either ‘indeterminate’ or ‘inconclusive’.
Line 47: include the number of HCWs who had all negative results.
Line 51: the use of the word ‘around’ is inappropriate – remove.
Line 55: remove ‘yet not all’
Introduction:
Line 64: ‘… (SARS-CoV-2) spread to Belgium, among other countries,...’
Line 65: ‘travelers returning from Tuscany…”
Line 69: ‘… were confined to their homes’
Line 72: remove first word “has”
Line 76: ‘.. nosocomial outbreaks were considered to..’
Line 80: remove ‘In fact”
Line 81: remove “thereby suggesting”, replace with “and”
Line 82: ‘.. other studies pointed to..’
Materials and Methods:
Line 110: (220/117) – what does this relate to/refer to? Can you reference the study please.
Line 122: remove the word “Few”
Lines 123 and 126: change the word ‘doubtful’ to either ‘indeterminate’ or ‘inconclusive’
Line 129: change to ‘Sequencing of SARS-CoV-2 genomes was performed for 45 samples’
Line 133: change to VILO TM, also please reference supplier. Change to H2O.
Line 138: reference for ARTIC Network sequencing protocol.
Line 175: replace ‘inferior to’ with ‘less than’
Results:
Line 194-195: You have stated that the cohort was skewed towards females, with only 20% comprising males. Could you comment on this?
Line 240 and 242 – the word ‘around’ is used twice and is not appropriate scientifically. It suggests that the numbers are not exact. For example – ‘around five pairs of samples seem to be directly related to each other’. Using the word ‘around’ suggests that the number of pairs could be four, or it could be six. Suggest rewording these sentences to use more concise language.
Line 254: remove the hyphen in ‘period’
Figure 4: This figure is very dense and difficult to read properly. It is hard to define the clades represented by the red nodes. Could you extract/zoom into the clades for the 35 transmission events and include them in the figure?
Discussion:
Line 291: remove e.g. and replace with ‘for example’
Line 311: remove the word ‘be’
Line 312: remove hyphen in outbreaks
Line 316 and 317: please define the HCWs at higher risk of being infected – such as nurses and paramedics?
Line 332 and 333: remove ‘most prone to performed voluntary RT-PCR testing’ and replace with ‘demonstrating higher attendance rates’
Conclusion:
Line 355: ‘during the countries first and second’
Line 356: remove words ‘in Belgium”
Round 2
Reviewer 1 Report
The authors have addressed my comments. This paper can be be published in the current form.